# The Importance of Molecular Structure for Textural and Physicochemical Properties of Extruded Wheat Flour

**DOI:** 10.3390/foods14101829

**Published:** 2025-05-21

**Authors:** Yuan Chai, Ruibin Wang, Bo Zhang, Yonglu Tang, Chaosu Li, Boli Guo, Ming Li

**Affiliations:** 1College of Food Science and Engineering, Shanxi Agricultural University, Jinzhong 030801, China; chaiyuan2027@163.com; 2Institute of Food Science and Technology, Chinese Academy of Agriculture Sciences/Comprehensive Utilization Laboratory of Cereal and Oil Processing, Ministry of Agriculture and Rural, Beijing 100193, China; zjzb1978@126.com; 3College of Food Science and Technology, Hebei Agricultural University, Baoding 071051, China; wangruibinsx@163.com; 4Crop Research Institute of Sichuan Academy of Agricultural Sciences, Environment-Friendly Crop Germplasm Innovation and Genetic Improvement Key Laboratory of Sichuan Province, Chengdu 610066, China; ttyycc88@163.com (Y.T.); xiaoli1755@163.com (C.L.)

**Keywords:** extrusion expansion, degradation, molecular structure, temperature, specific mechanical energy

## Abstract

This study elucidated the mechanistic interplay between the extrusion parameters (temperature and screw speed), starch molecular architecture (chain-length distribution), and key physicochemical properties of wheat flour extrudates. Four wheat flours with varied amylose contents were extruded, where the average hydrodynamic radius ($\overset{-}{R_{h}}$) was reduced by 75.5% in normal wheat (e.g., CM55), while waxy wheat (WW) exhibited higher $\overset{-}{R_{h}}$. Crispness correlated negatively with long amylopectin branches (36 < X ≤ 100), with WW displaying superior crispness (12.22 N/mm). Short amylopectin chains (X 6–36) increased under thermomechanical stress, enhancing the expansion index (SEI), whereas long chains (X > 100) restricted expansion. Temperature may modulate color difference (Δ*E*) via Maillard reactions, while higher specific mechanical energy (SME) intensified browning. Higher temperatures (>170 °C), rather than SME, caused significant changes in the proportion of short branches and long branches, with SME exhibiting a negative correlation with $\overset{-}{R_{h}}$, indicative of substantial molecular degradation. The starch chain-length distribution, rather than amylose content alone, dictates extrudate functionality.

## 1. Introduction

Extrusion–expansion technology has revolutionized the snack food industry, extending its reach beyond conventional puffed snacks to encompass functional ingredient manufacturing, texturized plant proteins, and ready-to-eat cereals. Key physicochemical properties such as expansion ratio, crispness, water absorption, and solubility are vital to the quality of breakfast cereal products [1].

Raw material selection greatly affects extrusion outcomes, with starch amylose content altering melt viscosity [2] and influencing extrudate expansion [3]. Wheat was prone to greater susceptibility to heat and shear than corn grits when they are processed with a twin screw extruder [4]. Although corn extrudates can exhibit higher expansion than wheat extrudates under specific conditions, these properties are highly dependent on processing parameters. For example, at higher screw speeds, corn may expand less than wheat [5]. In addition, the moisture content, or rather water activity, is closely related to the crispness and hardness of extruded starch products. When the water activity reaches 0.31, the crispness of the food undergoes a sharp decline [6]. The higher feed moisture content of rice-based expanded snacks leads to denser, less expanded products with a higher water absorption index (WAI), increased hardness, and reduced crispness [7]; thus, low moisture extrusion is favored when an increase in expansion and crispness is desired.

Thermal energy and mechanical energy are the primary drivers of structural changes in extrudates [8,9]. The extrusion temperature plays a pivotal role in dictating the viscosity of the extruded melt, which directly impacts the product’s expansion volume and hardness [10,11,12]. A temperature rise from 120 °C to 180 °C fosters molten starch and bubble formation, thereby boosting the expansion rate; however, temperatures above 180 °C can result in over-expansion and structural collapse [11].

Extrusion can change the multiscale structures of starch, including granular gelatinization, and molecular degradation, which further improves the cooking, texture, and eating qualities of noodles [13]. Generally, screw speed plays a crucial role in the degradation of starch [14,15]. Li et al. [14] found that excessive depolymerization was caused by specific mechanical energy (SME) rather than thermal degradation. The interplay between extrusion temperature and SME exerts a complex influence on product structure. The extent to which individual factors dominate the impact on the extrudates remains unclear. In addition, the correlation between the extrusion parameters, the fine molecular structure of starch, and the physicochemical properties of the final product is not consistent. This discrepancy primarily stems from an insufficient understanding of the structural changes in materials during extrusion.

Although pure starch has been commonly used in extrusion model systems [16,17], wheat flour was selected as the primary material in this study to more closely mimic industrial snack production processes. Wheat flour, containing starch, proteins, and other minor components, provides a complex matrix that reflects real-world extrusion scenarios. The presence of gluten and other flour components may influence the rheological properties of the extrusion melt [18] and interact with starch during processing, which could allow us to investigate the variations in starch composition within a more complex matrix.

Hence, this study analyzed the physicochemical attributes of wheat flour extrudates with distinct amylose content by controlling the mechanical or thermal energy input during extrusion. The objectives were to (1) identify wheat varieties that are best suited for the production of high-quality extruded foods, (2) investigate the impact of processing parameters on the starch molecular structure of extrudates, and (3) establish a correlation between the starch molecular structure and the physicochemical properties. The interplay between SME and temperature on the starch degradation and physicochemical properties was also discussed, offering valuable guidance for optimizing cereal extrusion processes.

## 2. Materials and Methods

### 2.1. Materials

Raw materials: Three wheat varieties (CM 66, CM 55, and CM 104) were gifted from the Sichuan Academy of Agricultural Sciences (Chengdu, China). A waxy wheat variety (Tiannuo 693, WW) was provided by the Institute of Genetics and Developmental Biology, Chinese Academy of Sciences (Beijing, China). Wheat grains were ground by a laboratory Buhler mill (MLU 202, BUHLER, Uzwil, Switzerland) and passed through an 80-mesh sieve (177 µm) (70% yield). The protein, damaged starch, lipid, and ash contents of different wheat varieties were determined by the AACC methods AACC 56–10, AACC 76–33.01, AACC 30–25, and AACC 08–01. The starch content was measured using the total starch analysis kits from Megazyme International (Bray, Ireland). All chemicals used were of analytical grade.

Wheat starches from different flours were extracted using a handwashing method following the methods of Zou et al. [19]. Briefly, wheat flour (150 g) was mixed with 80 mL of distilled water to form a dough, and the dough was washed with distilled water until no more starch was released. The starch slurry was centrifuged at 4000× *g* for 10 min to remove the supernatant, and then the protein layer on the surface was scraped off with a spatula. This washing cycle was repeated three times to remove residual protein. The starch was lyophilized with a freeze dryer (ALPHA 1-2 LD plus, CHRIST, Osterode, Germany) and ground to pass through a 100-mesh sieve (149 µm). The starch isolation yield was 88.95% with a purity of 95.22%.

### 2.2. Extrusion

Wheat extrudates were prepared using a twin-screw extruder (DSE-25, Brabender, Duisburg, Germany). The ratio of screw length to diameter (L/D) is 20:1 and the diameter of the die is 5 mm. Extrusion was performed under the barrel temperature (150–190 °C) (Zone V) and screw speed (100–190 rpm). Different varieties of wheat flour were pre-conditioned to a moisture content of 16%, followed by equilibration at 4 °C overnight before extrusion, and the feed rate was 35 g/min (Table 1). The system parameters, such as torque and SME during extrusion, were monitored and calculated following Equation (1) [20]. Extrudates were freeze-dried, ground using a high-speed rotary grinder (ZM200, Retsch, Leipzig, Germany), and sieved through a 70-mesh sieve (212 µm) for subsequent analysis. The extrusion trials were replicated. The repeatability of the experiment trials was verified by repeating one of the trials, and the samples were collected and analyzed. The temperature settings for the five zones of the extruder are shown in Appendix A.(1)SEM (W·h/kg)=(2π×n×T)/F
where *n* is the screw speed (rpm), *T* is the torque (N·m), and *F* is the feed rate (g/min).

### 2.3. Fine Molecular Structure of Starch

Native wheat starch (extracted by the handwashing method in Section 2.1) and wheat flour extrudates were dissolved in DMSO (ACS grade, Merck, Darmstadt, Germany) with 0.5% *w*/*w* LiBr (Reagent Plus) for 8 h at 80 °C, and then characterized using a size exclusion chromatography (SEC) system (Agilent Technologies, Waldbronn, Germany) equipped with GRAM 30 and 3000 analytical columns (PSS) and a refractive index (RI) detector (RID-10 A, Shimadzu Corp, Kyoto, Japan) following the methods of Vilaplana and Gilbert [21]. The debranched structures were determined using the same SEC system but equipped with GRAM 100 and 1000 analytical columns (PSS).

The size distributions of whole starch molecules were plotted as the weight distribution *w*_br_(log*R*_h_) against the equivalent hydrodynamic radius (*R_h_*). All SEC molecular size distributions were normalized to the maximum peak height; the average *R_h_* values ($\overset{-}{R_{h}}$) were obtained according to the description of Vilaplana and Gilbert [21].

The SEC weight distribution *w*_de_(log*R*_h_) of the debranched starches was plotted relative to *R*_h_ or degree of polymerization (DP), *X*. The distributions were normalized to the highest amylopectin peak. The chains with (DP), *X* smaller than 100 were normally assigned as amylopectin (Ap) chains, while those with (DP), *X* above 100 were amylose (Am) chains. For a linear polymer (such as debranched starch), the number distribution of chains (obtained by debranching) is related to the corresponding SEC weight distribution by Castro et al. [22]. The amylose content was calculated as the area under the curve (AUC) of Am branches (X ≥ 100) divided by the total area of all Ap and Am branches [23]; the relative content of amylopectin is determined by dividing the AUC of debranched Ap fragments by the total AUC of all Ap and Am branches.

### 2.4. Physicochemical Properties of Extrudates

#### 2.4.1. Sectional Expansion Index (SEI) and Apparent Density

The sectional expansion index of extrudates was calculated by dividing the cross-sectional diameter by the diameter of the die (5 mm) [24]. The cross-sectional diameters of extrudates were measured using a digital vernier caliper (Guilin Guanglu Measuring Instrument Co., Ltd., Guilin, China). Apparent density (*ρ*) was evaluated using the method described by Nascimento et al. [25].

#### 2.4.2. Water Absorption Index (WAI) and Water Solubility Index (WSI)

The WAI was defined as the wet pellet weight per gram of sample, whereas the WSI was expressed as the percentage of dried supernatant solids relative to the original sample mass. The WAI and WSI of ground powder were assessed using the method described by Acharya et al. [26]. In brief, 2.0 g of finely powdered sample was suspended in 25 mL of distilled water in a pre-weighed centrifuge tube and mixed to homogeneity. Following incubation in a water bath at 30 °C for 30 min with intermittent shaking at 10 min intervals, the suspension was centrifuged at 4200× *g* for 15 min. The resultant supernatant was oven-dried at 105 °C to constant weight. The WAI and WSI were derived from Equations (2) and (3), respectively:(2)$$\mathrm{WAI}=\frac{m_{b}}{m_{a}}$$(3)$$\mathrm{WSI}=\frac{m_{c}}{m_{a}}\times 100\%$$
where *m_a_*, *m_b_*, and *m_c_* are the weights of the initial sample, wet pellets, or supernatant dry solids, respectively.

#### 2.4.3. Color

The color of the wheat flour before and after extrusion was analyzed using a colorimeter (CR-400, Konica Minolta, Tokyo, Japan) [27]. The CIELAB color system quantifies color in terms of lightness (*L*∗: 0 for black, 100 for white), red/green chromaticity (*a*∗: −*a*∗ for green, +*a*∗ for red), and yellow/blue chromaticity (*b*∗: −*b*∗ for blue, +*b*∗ for yellow), and the total color difference (Δ*E*) indicates the color difference from the standard plate calculated as in Equation (4):(4)$$\Delta E=\sqrt{{\left(L^{*}-L_{S}^{*}\right)}^{2}+{\left(a^{*}-a_{S}^{*}\right)}^{2}+{\left(b^{*}-b_{S}^{*}\right)}^{2}}$$
where *L_s_**, *a_s_**, and *b_s_** are the standard values from a white tile, which were 97.13, 0.21, and 1.87, respectively.

#### 2.4.4. Hardness and Crispness

Textural properties were determined according to the method of Altan et al. [28] with slight modifications. Extrudates were freeze-dried and cut into 5 cm pieces for textural analysis. The hardness and crispness of extrudates were analyzed using a texture analyzer (TA-XT2i, Stable Micro Systems, Godalming, UK) equipped with an HDP/KS5 Kramer shear cell (5-blade configuration) probe. The load cell was 25 kg. Testing parameters included pre-test, test, and post-test speeds of 2.00 mm/s, 2.00 mm/s, and 10.00 mm/s, respectively, with a trigger force of 10 g and a compression distance of 48 mm. Textural parameters such as peak force and slope were obtained from the force displacement graph, evaluated as hardness (N) and crispness (N/mm), respectively.

### 2.5. Statistical Analysis

All the determinations were conducted in triplicate unless otherwise stated. All data results were presented as mean (M) ± standard deviation (SD). The significant differences were determined by Duncan’s multiple range test (*p* < 0.05), and Pearson correlations were analyzed by SPSS Statistics 26 (SPSS Inc., Chicago, IL, USA).

## 3. Results

### 3.1. Proximate Composition Analysis

The wheat variety directly affects the eating quality of wheat extrudates owing to their chemical composition. Specifically, regarding wheat gluten, premium low-gluten wheat yields a crispy texture when subjected to extrusion processing, whereas high-gluten wheat develops an undesirably hard and dense texture under the same conditions. CM55, CM66, and CM104 are the main wheat varieties in Sichuan, known for their premium low-gluten content. Among these, CM 66 exhibited the lowest extrusion hardness; therefore, it was selected as the raw material to study the effects of different processing conditions on the physicochemical properties of extrudates. Besides protein, the starch molecular structure also influences the texture of extrudates, which drives the specific selection of waxy wheat (WW) cultivars.

The proximate compositions (dry basis) of the wheat flours from different varieties are presented in Table 2. Notably, CM66 contained a relatively higher amount of amylose compared to other samples. The protein content of CM55, CM66, and CM104 was similar and was significantly lower than that of WW. WW, on the other hand, exhibited significantly higher protein content and WSI and lower amylose content than other varieties. CM104 showed the lowest protein content. Except for CM55, no significant differences were observed in the WAI among the samples.

### 3.2. SME with Varied Extrusion Conditions and Wheat Varieties

SME quantifies the shear and extrusion resistance a material offers during expansion [29]. When different wheat varieties were extruded under the same extrusion conditions, WW exhibited the lowest SME (396.60 W·h/kg), significantly less than the other samples, with CM104 showing the highest SME (660.24 W·h/kg). This can be explained by the lower amylose content of WW (Table 3), which is consistent with a previous study [14]. However, other factors, such as the fine molecular structure of starch or non-starch components (e.g., proteins and lipids), might explain the distinct SME values observed for CM55, CM66, and CM104, given their similar amylose content and identical extrusion conditions.

For CM66, increasing the screw speed from 100 to 190 rpm (at a constant barrel temperature of 170 °C) significantly raised the SME from 326.51 to 604.74 W·h/kg (Table 1). This increase is attributed to the synergistic effects of intensified mechanical shearing and starch structural degradation. High screw speeds amplify barrel shear forces, causing starch chain scission, enhancing starch degradation [30].

Additionally, barrel temperature significantly affected the SME of CM66, as elevated temperatures reduced the melt viscosity and resistance [20]. Lower temperatures (e.g., 150 °C) yielded lower specific energy consumption, while temperatures above 170 °C produced similar SME values, reflecting a balance between thermal softening and shear effects.

### 3.3. Changes in the Molecular Structure of Wheat Flour Extrudates and Wheat Starch

#### 3.3.1. Molecular Structure of Native Starch from Different Wheat Varieties

The molecular structure of the native starch from different wheat varieties was characterized using typical SEC weight distributions of whole branched (*w*_br_(log*R*_h_)) and debranched starch (*w*_de_(log*R*_h_)) (Table 3 and Figure 1a,b). Native starch from CM55 exhibited the highest amylose content (31.08%), whereas WW consisted of the lowest amylose content (3.01%). The amylose content in CM66 and CM104 was similar (~29%), but significantly lower than that in CM55 and higher than that in WW. A greater proportion of amylopectin in native starch from WW contributed to a higher $\overset{-}{R_{h}}$ (102.35 nm) than that of other wheat starch (84.40–86.66 nm). A higher proportion of chains with 12 < *X* ≤ 24 compared to other chains was observed in debranched starch, where WW had the highest proportion (37.95%), and CM55 showed shorter chains (6 < *X* ≤ 24).

#### 3.3.2. Changes in Starch Molecular Structure of Different Wheat Flour Varieties Under Extrusion

As shown in Table 3 and Figure 1a,b, extrusion at 170 °C and 190 rpm induced substantial starch molecular degradation across all wheat varieties $\overset{-}{R_{h}}$ values exhibited marked reductions, with the CM55 demonstrating the highest degradation susceptibility (85.37–20.94 nm). The degradation susceptibility followed the pattern CM55 > CM104 > CM66 > WW. Common wheat starch displayed molecular diameter reductions equivalent to 23.27–26.09% of the initial values. In contrast, the WW retained higher structural integrity, with its molecular radius reduced to 47.59% of the original. This likely correlates with WW’s elevated amylopectin content. This differential degradation behavior aligns with the SME variations, where higher amylopectin content corresponded to reduced SME input requirements (Table 1).

Furthermore, compared to native starches, thermomechanical processing selectively altered the amylopectin chain length distribution. The proportions of chains in the ranges 12 < *X* ≤ 24 and 36 < *X* ≤ 100 exhibited notable reductions (Table 3). This redistribution likely reflects preferential degradation of longer amylopectin chains during extrusion, with inter-varietal differences in branching patterns (e.g., 24 < *X* ≤ 36/36 < *X* ≤ 100 chain ratios) amplifying this differential degradation.

Different from previous studies [14,31], there was an unusual increase in the *X* > 100 range of long-chain molecules (from 3.01–31.08% to 19.47–61.97%). This phenomenon may be attributed to the intensified interactions between starch and proteins during high-temperature extrusion, where the formation of starch–starch or starch–protein complexes could alter the molecular weight distribution of starch, consequently leading to a modification in the proportion of molecules within the *X* > 100 range.

#### 3.3.3. Changes in Starch Molecular Structure of CM 66 Under Different Extrusion Conditions

The impact of temperature on molecular degradation was found to be significantly greater than that of shear (Table 3 and Figure 1c–f). Specifically, as the temperature rose from 150 °C to 190 °C with a constant screw speed of 190 rpm, there was a marked decrease in the proportion of medium and long chain amylopectin molecules. The proportion of molecules within the 12 ≤ *X* ≤ 24 intervals decreased from 20.98% at 150 °C to 18.63% at 170 °C, and finally to 11.67% at 190 °C, representing a decrease of 44.4%. The proportion of molecules in the 24 ≤ *X* ≤ 36 interval decreased from 9.44% at 150 °C to 8.96% at 170 °C, and then to 5.20% at 190 °C, amounting to a 45% decrease. Elevated temperatures intensify the fragmentation of molecular chains. When the temperature was fixed at 170 °C, the increase in screw speed from 100 rpm to 190 rpm also enhanced the mechanical degradation of medium and long chain molecules (Table 3); however, this increase due to shear contributed less to molecular chain fragmentation compared to the temperature effect described above (Table 4).

To further analyze the changes within the amylopectin internal structure, the proportion of the 6 < *X* ≤ 12 interval by AP (amylopectin content) was calculated to eliminate the influence of fluctuations in the total amylopectin content. The absolute proportion of the 6 < *X* ≤ 12 chains of CM66 extruded at 170 °C/190 rpm was 9.57% (AP 51.2%), yielding a relative proportion of 18.7%. The relative proportions were 21.1% at 150 °C/190 rpm and 19.7% at 190 °C/190 rpm. At the highest temperature (190 °C), although the absolute proportion of the chains (6 < *X* ≤ 12) decreased, the relative proportion only slightly declined due to the substantial reduction in AP. The degradation rate of short chains was similar to that of total amylopectin, and the proportion of short chains in the residual amylopectin did not change significantly.

Prior studies [14] have demonstrated that the proportion of amylopectin chains remained largely unchanged across varying screw speeds and extrusion temperatures. However, this study reveals that thermo-mechanical coupling exerts directional chain recombination. Under processing conditions of 170 °C/190 rpm, the proportion of long-chain amylopectin segments (36 < *X* ≤ 100) altered to 11.75%, representing an 18.57% absolute increase compared to the 170 °C/100 rpm (9.91%). Yet, under the extreme condition (190 °C/190 rpm), the AP ratio dramatically dropped to 32.2%, with *X* > 100 chain lengths accounting for 67.8%, necessitating further validation to determine whether this reflects a genuine increase in long chains or an artifact of detection.

### 3.4. The Impact of Extrusion Parameters on the Physicochemical Properties of Extrudates

#### 3.4.1. SEI, Apparent Density, Hardness, and Crispness

The SEI of wheat flour extrudates ranged from 1.59 to 3.64 (Table 1). For different wheat varieties under the same extrusion conditions, CM55 exhibited the highest SEI, followed sequentially by CM66, WW, and CM104 (Figure 2a,b). The apparent densities (AD) fluctuated between 0.075 and 0.329 g/cm^3^. Among the different wheat varieties, CM66 demonstrated the lowest AD under extrusion at 170 °C and 190 rpm. Notably, the AD of WW extrudates was higher than that of the other samples processed under the same extrusion conditions. Under the same extrusion parameters, the WW extrudates showed the highest hardness (84.17 N), whereas CM66 exhibited the lowest value (34.92 N). For CM66 extrudates, a higher screw speed resulted in a lower hardness, attributed to shear-induced starch degradation. The ECM66-170-100 extrudate exhibited significantly higher hardness (69.36 N) compared to other CM66 extrudates. This is primarily attributed to its unique molecular structure, formed under low SME (326.51 W·h/kg). The low SME reduced starch chain breakage, as evidenced by the ECM66-170-100 sample showing a high $\overset{-}{R_{h}}$. The less extensive chain degradation and a denser network may significantly increase hardness. There was no significant change in hardness from 160 to 190 rpm. Hardness exhibited a non-linear dependence on extrusion temperature, reaching its minimum at 170 °C with elevated values at both 150 °C and 190 °C.

The crispness, an important textural attribute determining the acceptability of extrudates, was quantified by the slope or the number of peaks. Under the same extrusion parameters, the WW extrudate exhibited the highest crispness (12.22 N/mm), while other varieties showed similar crispness (5.56–7.43 N/mm). For CM66 extrudates, lower screw speeds tend to result in higher crispness (9.11 N/mm at 100 rpm), whereas higher speeds, lead to inverse results (such as 5.56 N/mm at 190 rpm). Conversely, temperature caused the crispness to have an opposite tendency (5.65 N/mm at 150 °C vs 9.52 N/mm at 190 °C).

The equilibrium between hardness and crispness of the extruded product is important [24]. Based on sensory evaluation, ECM66-170-190 emerged as the preferred sample in this study compared to other wheat varieties. It demonstrated the lowest hardness (34.92 N) and a comparatively low crispness (5.65 N/mm) consistent with consumer preferences for expanded snacks.

#### 3.4.2. WAI and WSI

The WAI refers to the ability of a macromolecule to absorb and retain moisture. A high WAI endows extrudate with a faster softening effect, resulting from absorbing moisture more quickly. The WAI of extruded wheat flour products ranged from 0.91 to 1.66 (Table 5). The wheat varieties significantly interfered with the WAI of extrudates, with CM104 exhibiting the highest value (1.65) and CM66 showing the lowest (0.91). For CM66, a higher screw speed resulted in a lower WAI. The WAI value at 170 °C was comparatively lower than those at 150 °C and 190 °C. Notably, while it is widely accepted that extrusion typically increases WAI compared to raw materials, the opposite trend observed here may be attributed to the relatively extreme processing conditions employed in this study: high temperatures (150–190 °C) and low moisture content (16%). These conditions could lead to amylopectin degradation and complexation between starch and other macromolecules, thereby restricting moisture absorption. Additionally, the use of ground extrudate powders with higher particle size (lower specific surface area) further contributed to the reduced WAI.

The WSI reflects increased leaching of low-molecular-weight solutes (e.g., dextrins <*X* 15), potentially compromising structural integrity through matrix discontinuity. Under the same extrusion condition, CM66 exhibited the highest WSI (89.23%), while CM104 showed a slightly lower WSI (80.93%). When the screw speed increased from 100 to 190 rpm, the WSI increased by 9.75% ranging from 81.30% to 89.23% (*p* < 0.001), confirming that shear degradation predominantly enhanced solubility. The WSI values at higher temperatures (such as 190 °C) were lower, possibly due to caramelization reactions and the formation of melanoidins at high temperatures increasing the proportion of insoluble high molecular weight polymers.

#### 3.4.3. Color

The Δ*E* quantitatively reflects the extent of color deviation in extrudates, with higher values indicating darker coloration. The color of extrudates from different wheat varieties was significantly different. Specifically, CM66 exhibited the most pronounced Δ*E* (34.77), 1.7-fold higher than WW (20.33) (Table 5). Notably, WW showed the highest brightness (82.68), whereas CM55 exhibited the highest Δ*E*, *a**, and *b**. This chromatic variation may be due to starch depolymerization, generating reducing dextrins, while Maillard reactions between liberated sugars and glutamine residues produce melanoidins and 5-hydroxymethylfurfural (HMF) derivatives [32]. The Δ*E* decreased from 35.92 at 150 °C to 27.78 at 190 °C but increased as the screw speed rose (a maximum of 34.77 at 190 rpm) (Table 5). High shear forces or temperature could increase the amount of hydroxymethyl furfural (HMF) [33].

The Δ*E* decreased from 35.92 at 150 °C to 27.78 at 190 °C but increased as the screw speed rose (a maximum of 34.77 at 190 rpm) (Table 5). Similarly, *L** increased by 14%, ranging from 66.96–76.33 with rising temperature (150–190 °C). This enhancement in brightness occurred despite the Maillard reaction generating browning products (contributing to Δ*E*), as the dominant moisture flash-off (>8% w.b. reduction) at elevated temperatures enhanced light reflection. Conversely, *L** decreased by 10.6% (77.66–69.41) with screw speed elevation (100–190 rpm). Differently, *a** decreased when temperatures exceeded 170 °C (4.77 at 170 °C vs. 3.23 at 190 °C), coinciding with the polymerization of Maillard intermediates into high-molecular-weight melanoidins (>10^4^ Da) [34], which shifted chromaticity toward neutral tones. The redness (*a**) increased significantly with the rise of screw speed. *b** peaked at 170 °C and 190 rpm (22.36) and ranged from 18.08 to 21.14 at other rotational speeds. However, the reason for the formation of yellowness remains unknown.

### 3.5. Selection of Different Wheat Varieties

The selection of raw materials for breakfast cereals is of paramount importance, as raw materials with larger molecular sizes ($\overset{-}{R_{h}}$, waxy wheat in this case) could be chosen to enhance product crispness. The crispness of extrudates from different wheat varieties exhibited a positive correlation with the average molecular size ($\overset{-}{R_{h}}$). This suggests that when selecting breakfast cereal raw materials, crispness may be more dependent on the inherent properties of the material, and that starch with larger molecular sizes may produce products with greater crispness. Breakfast cereal products with a higher level of crispness could also exhibit a more consistent and potentially lighter color.

For wheat flour derived from different varieties, no significant correlation was observed between the processing parameters, molecular structure, and the hardness of the extruded products (Table 6). High-temperature, high-shear extrusion simultaneously triggers starch degradation [35] and promotes the Maillard reaction [36], disrupting the structure of the extrudate and leading to a significant decrease in hardness and crispness. This is in contrast to the findings of [37], in which it is observed that extrudates made from waxy barley flours were softer than those from regular barley flours. Additionally, that study indicated that the apparent modulus of compression of waxy barley extrudates might increase under varying processing conditions, suggesting that changes in processing temperatures or screw speeds can affect the material’s resistance to deformation.

The molecular size of wheat starch from different wheat varieties is inversely related to torque and SME (Table 6). A decrease in torque and SME leads to an increase in $\overset{-}{R_{h}}$. No significant correlations were observed between the WSI, WAI, and SEI concerning processing parameters or molecular structure. It is noted that the WAI and WSI were measured using grounded extrudate powders, while the SEI was measured using extrudates in their original form. This may interfere with their correlation. Unlike the effects of temperature or screw speed on WSI and WAI, the choice of raw materials does not appear to be a critical determinant for these characteristics.

The molecular structure of starch, within a more complex matrix, varied greatly (Figure 1). Future studies are suggested to complement these findings by using pure starches with or without protein in a more controlled environment to elucidate the specific contributions of protein to starch molecular structure changes, which will offer some more practical insights for industrial applications.

### 3.6. Extrusion Parameter Modulation for Better Physicochemical Properties of Wheat Flour Extrudates

The physicochemical properties of the extrudates from CM66 were influenced greatly by the extrusion screw speed (Table 4). In terms of crispness, a negative correlation was observed with the fraction of long branches in amylopectin (36 < *X* ≤ 100). A higher proportion of long-chain branches may increase the viscosity of the extrudates, making them less prone to forming brittle structures during processing. As Reyniers, De Brier et al. [38] have noted, the content of short extractable amylose chains in potato flakes influences amylose crystal formation, expansion, and subsequent oil absorption during deep-frying. Nevertheless, additional investigation is necessary to validate the proposed hypothesis. However, the hardness of CM66 was insignificantly affected by processing parameters and molecular structure.

WSI of CM66 was positively correlated with screw speed, which is consistent with [5]. Conversely, WAI was negatively correlated with screw speed and SME. Higher screw speed and SME could reduce water absorption. Mechanical shear could degrade the molecular structure (Table 4), leading to the exposure of more water-soluble constituents (such as starch molecules) and thereby increasing water solubility [39]. From Table 4, the WAI is positively related to the short chains of amylopectin, while WSI is negatively related to those chains. However, it is difficult to understand why the small chains could absorb water rather than dissolve in water. Taking small beads (short chains) and large chains (long chains) as an example, the small beads were easily surrounded and absorbed water, but they were not easily dispersed in water. The large chains, on the other hand, could more easily unfold and disperse in water, thus being more readily soluble. Still, this remains to be explained.

Comparing the extrusion of CM66 under conditions of 170 °C/190 rpm and 190 °C/190 rpm, similar SME values, approximately 610 W⋅h/kg, were observed (Table 1). However, the increase in temperature led to greater degradation, as indicated by the reduction in $\overset{-}{R_{h}}$ from 23.21 nm to 19.64 nm (Table 3), and the extrudates exhibited higher hardness and brittleness. This finding contrasts with previous studies [14,31], which suggested that temperature has a less significant effect on molecular degradation compared to SME. The CM66 starch used in this study may be more sensitive to temperature changes than maize starches used in previous research, leading to more pronounced degradation at higher temperatures. Or SME values in both conditions might have reached a threshold for CM 66, where further increases do not significantly contribute to degradation, and thus temperature becomes a more dominant factor. More work can be carried out in the future in interpreting the differences in degradation to control the extrusion process more effectively.

Based on the results of this study, reducing torque appropriately may decrease the water absorption and improve the lightness of CM66. Color change (Δ*E*) was negatively associated with torque, and *b** was positively correlated with SME. Mechanical shear and temperature affect light scattering and absorption by altering the molecular structure (such as the degree of degradation), thereby influencing color. Additionally, the Maillard reaction during extrusion is a result of the combined effects of mechanical degradation and temperature. Understanding this mechanism can aid in the precise control of the color of extrudates.

In addition, the apparent density of CM66 extrudates was significantly influenced by the processing parameters and the starch molecular structure. The apparent density exhibited a significant positive correlation with AM and a significant negative correlation with short-to-medium-branched starch (6 < X ≤ 36). The higher content of short-to-medium-branched starch (6 < X ≤ 36) might enhance expansion through improved chain entanglement and gas retention during extrusion, thereby reducing density.

### 3.7. Screw Speed and Temperature Effects on the Texture of Extruded Grains: Hardness and Crispness

Texture is a primary parameter in extruded cereal production, influenced by both raw material composition and processing conditions [40]. In this study, an inverse correlation was observed between the hardness of CM66 extrudates and the rate of screw speed (Figure 3). Similarly, research has demonstrated that the hardness of corn- and barley-based extrudates also decreases with increasing screw speed [41,42]. The hardness of CM66 extrudates was significantly greater at elevated temperatures (190 °C) in comparison to lower temperatures. Conversely, the hardness of composite extruded products made from sorghum, barley, and chickpeas decreases with increasing extrusion temperature [43]; increased barrel temperature lowers melt viscosity, which encourages bubble growth, reducing cell wall thickness during extrusion and lowering the hardness; and the hardness of chickpea flour-based snacks decreases with increasing extrusion temperature due to the high correlation with bulk density [29]. The change in the crispness of CM66 extrudates shows the same trend as hardness. While Ding et al. [7] found that an increase in temperature led to a subsequent increase in the crispness of rice-based expanded snacks, their study emphasized that moisture content was the primary factor, whereas our results (with controlled moisture) suggest temperature-driven mechanisms may dominate in wheat systems.

## 4. Conclusions

CM66 extrudates showed the lowest hardness and crispness compared to other wheat varieties, demonstrating superior hydro-solubility characteristics, rendering this cultivar optimally suited for breakfast cereal production. These physicochemical properties may be attributable to the fine molecular structure of starch induced by extrusion temperature and screw speed. The specialized wheat varieties with different chain length distributions of starch should be cultivated for extrudates. Instead, waxy wheat presented the least degradation sensitivity induced by high temperature and screw speed, contributing the control of crispness. In addition, the wheat materials used in this study had different gluten strengths. Therefore, the observed unexpected increase in long-chain molecules (X > 100) is speculated to result from starch–protein (especially gluten) interactions under extreme conditions, which requires further clarification.

## Figures and Tables

**Figure 1 foods-14-01829-f001:**
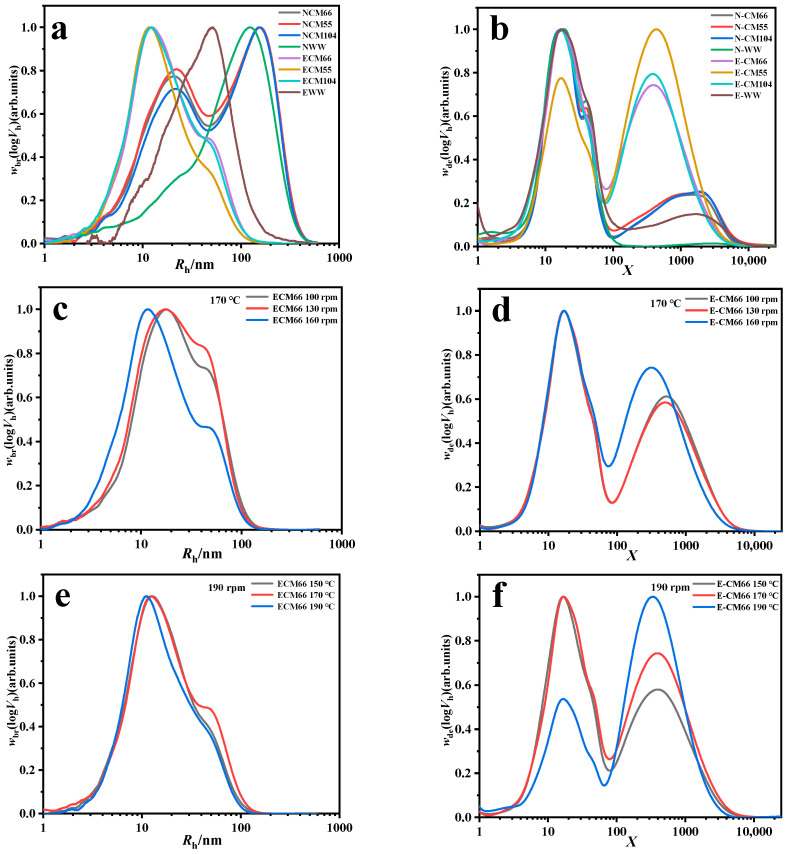
SEC weight distributions of branched and debranched starch molecules. NCM66, NCM55, NCM104, and NWW denote non-extruded wheat starch; ECM66, ECM55, ECM104, and EWW denote extruded wheat flours. (**a**,**b**) Weight distribution of branched and debranched wheat flour from different varieties before and after extrusion. (**c**,**d**) Weight distribution of branched and debranched CM66 flour under varying screw speeds. (**e**,**f**) Weight distribution of branched and debranched CM66 flour under different extrusion temperatures.

**Figure 2 foods-14-01829-f002:**
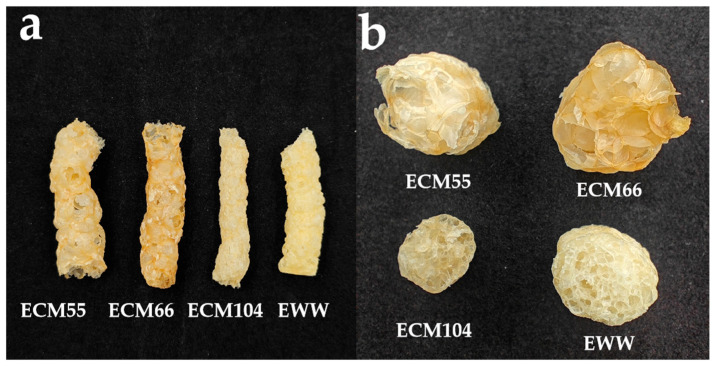
Strands (**a**) and cross-section morphology (**b**) of extrudates from different wheat varieties; ECM66, ECM55, ECM104, and EWW denote wheat extrudates obtained at 170 °C and 190 rpm.

**Figure 3 foods-14-01829-f003:**
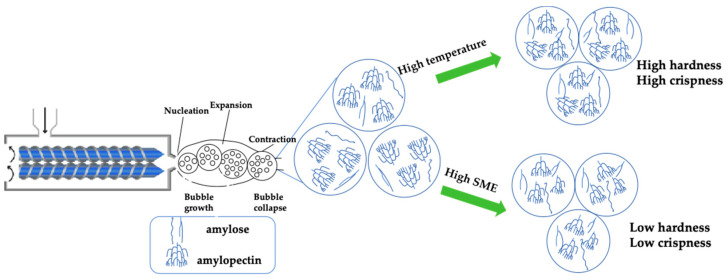
Schematic diagram of extrusion effects on hardness and brittleness of extrudates.

**Table 1 foods-14-01829-t001:** Extrusion response and textural properties of extruded wheat products with varied wheat variety, temperature, and screw speed.

Sample	T /°C	SS /rpm	MC /%	SME /W·h/kg	SEI	AD /g/cm^3^	Hardness /N	Crispness /N/mm
ECM66-170-100	170	100	16	326.51 ± 6.53 ^g^	3.64 ± 0.30 ^a^	0.120 ± 0.031 ^cd^	69.36 ± 8.33 ^b^	9.11 ± 4.05 ^bc^
ECM66-170-130	170	130	16	418.05 ± 7.37 ^e^	3.15 ± 0.24 ^b^	0.104 ± 0.013 ^cd^	58.8 ± 7.75 ^c^	8.40 ± 2.90 ^bcd^
ECM66-170-160	170	160	16	510.77 ± 7.48 ^d^	2.85 ± 0.22 ^d^	0.165 ± 0.027 ^b^	39.26 ± 3.92 ^e^	6.02 ± 1.50 ^cd^
ECM66-170-190	170	190	16	604.74 ± 27.18 ^b^	2.53 ± 0.17 ^e^	0.075 ± 0.008 ^d^	34.92 ± 5.21 ^e^	5.56 ± 1.08 ^d^
ECM66-150-190	150	190	16	576.52 ± 9.40 ^c^	3.16 ± 0.62 ^b^	0.103 ± 0.011 ^cd^	43.61 ± 12.2 ^de^	5.65 ± 1.37 ^d^
ECM66-190-190	190	190	16	617.05 ± 36.98 ^b^	1.59 ± 0.34 ^f^	0.329 ± 0.087 ^a^	54.69 ± 12.83 ^c^	9.52 ± 4.82 ^ab^
ECM55-170-190	170	190	16	603.51 ± 25.55 ^b^	3.05 ± 0.26 ^cd^	0.144 ± 0.023 ^bc^	44.66 ± 6.61 ^de^	6.16 ± 1.70 ^cd^
ECM104-170-190	170	190	16	660.24 ± 8.17 ^a^	1.78 ± 0.32 ^f^	0.120 ± 0.038 ^cd^	49.45 ± 6.44 ^cd^	7.43 ± 3.01 ^bcd^
EWW-170-190	170	190	16	396.60 ± 3.46 ^f^	2.36 ± 0.25 ^e^	0.166 ± 0.064 ^b^	84.17 ± 17.25 ^a^	12.22 ± 5.4 ^a^

Different letters in a column indicate significant differences (*p* < 0.05). ECM66, ECM55, ECM104, and EWW denote extruded wheat flours. T: temperature; SS: screw speed; MC: moisture content; SME: specific mechanical energy; SEI: sectional expansion index; AD: apparent density.

**Table 2 foods-14-01829-t002:** The chemical composition and functional properties of flours derived from different wheat varieties.

Wheat Variety	Protein /%	Starch /%	AM /%	Ai /%	Lipid /%	Ash /%	WAI	WSI /%
CM55	9.27 ± 0.01 ^b^	77.33 ± 1.13 ^b^	27.98 ± 0.91 ^b^	94.99 ± 0.16 ^a^	0.53 ± 0.01 ^a^	0.51 ± 0.02 ^a^	2.32 ± 0.01 ^a^	8.52 ± 0.15 ^b^
CM66	9.10 ± 0.26 ^b^	77.98 ± 0.49 ^ab^	31.08 ± 0.10 ^a^	92.86 ± 0.17 ^c^	0.31 ± 0.06 ^b^	0.43 ± 0.09 ^ab^	2.04 ± 0.01 ^b^	6.20 ± 0.05 ^c^
CM104	8.09 ± 0.13 ^c^	79.60 ± 0.27 ^a^	29.01 ± 0.68 ^b^	93.53 ± 0.22 ^b^	0.47 ± 0.05 ^a^	0.36 ± 0.04 ^b^	2.05 ± 0.02 ^b^	6.30 ± 0.05 ^c^
WW	13.1 ± 0.11 ^a^	72.44 ± 0.26 ^c^	3.01 ± 0.48 ^c^	70.36 ± 0.11 ^d^	0.21 ± 0.05 ^b^	0.40 ± 0.01 ^b^	2.03 ± 0.01 ^b^	10.82 ± 0.08 ^a^

Different letters in a column indicate significant differences (*p* < 0.05). AM: amylose content; Ai: iodine absorption rate of damaged starch; WAI: water absorption index; WSI: water solubility index.

**Table 3 foods-14-01829-t003:** Molecular structure of extruded wheat flour.

Samples	6 < X ≤ 12	12 < X ≤ 24	24 < X ≤ 36	36 < X ≤ 100	X > 100	$\overset{-}{R_{h}}$/nm
NCM66	14.65 ± 0.09 ^B^	30.05 ± 0.15 ^B^	12.34 ± 0.10 ^B^	12.87 ± 0.92 ^B^	27.98 ± 0.91 ^B^	84.40 ± 1.19 ^B^
NCM55	13.06 ± 0.25 ^C^	28.35 ± 0.22 ^C^	12.15 ± 0.41 ^B^	13.74 ± 0.41 ^B^	31.08 ± 0.10 ^A^	85.37 ± 2.27 ^B^
NCM104	14.23 ± 0.92 ^BC^	30.00 ± 0.44 ^B^	12.37 ± 0.02 ^B^	12.81 ± 0.22 ^B^	29.01 ± 0.68 ^B^	86.66 ± 0.61 ^B^
NWW	19.33 ± 0.43 ^A^	37.95 ± 0.07 ^A^	16.94 ± 0.12 ^A^	19.02 ± 0.27 ^A^	3.01 ± 0.48 ^C^	102.35 ± 0.26 ^A^
ECM66-170-100	11.59 ± 0.25 ^b^	21.02 ± 0.38 ^b^	9.38 ± 0.67 ^bcd^	9.91 ± 0.18 ^cd^	45.63 ± 0.92 ^e^	28.19 ± 0.52 ^b^
ECM66-170-130	11.30 ± 0.40 ^b^	21.37 ± 0.09 ^b^	9.85 ± 0.12 ^b^	10.21 ± 0.60 ^c^	44.56 ± 0.11 ^f^	26.77 ± 0.22 ^c^
ECM66-170-160	10.27 ± 0.57 ^cd^	19.02 ± 0.15 ^c^	8.77 ± 0.37 ^cd^	12.28 ± 0.45 ^b^	47.69 ± 0.21 ^d^	21.70 ± 0.36 ^ef^
ECM66-170-190	9.57 ± 0.18 ^d^	18.63 ± 0.10 ^c^	8.96 ± 0.04 ^bcd^	11.75 ± 0.13 ^b^	48.80 ± 0.48 ^c^	23.21 ± 0.21 ^d^
ECM66-150-190	11.91 ± 0.25 ^b^	20.98 ± 0.23 ^b^	9.44 ± 0.60 ^bc^	11.71 ± 0.61 ^b^	43.52 ± 0.19 ^f^	21.16 ± 0.21 ^f^
ECM66-190-190	6.34 ± 0.17 ^f^	11.67 ± 0.19 ^e^	5.20 ± 0.11 ^f^	7.36 ± 0.30 ^e^	67.80 ± 0.35 ^a^	19.64 ± 0.58 ^g^
ECM55-170-190	7.32 ± 0.55 ^e^	14.21 ± 0.08 ^d^	6.46 ± 0.33 ^e^	8.91 ± 0.70 ^d^	61.97 ± 0.54 ^b^	20.94 ± 0.33 ^f^
ECM104-170-190	10.45 ± 0.42 ^c^	18.85 ± 0.34 ^c^	8.48 ± 0.51 ^d^	10.18 ± 0.96 ^c^	49.79 ± 0.63 ^c^	22.61 ± 0.38 ^de^
EWW-170-190	15.61 ± 0.03 ^a^	28.85 ± 0.06 ^a^	14.49 ± 0.03 ^a^	17.36 ± 0.04 ^a^	19.47 ± 0.17 ^g^	48.71 ± 0.72 ^a^

Different letters in a column indicate significant differences (*p* < 0.05). Uppercase and lowercase letters denote untreated and extruded wheat flour samples, respectively. NCM66, NCM55, NCM104, and NWW denote non-extruded wheat starch; ECM66, ECM55, ECM104, and EWW denote extruded wheat flours. $\overset{-}{R_{h}}$: average hydrodynamic radius.

**Table 4 foods-14-01829-t004:** Correlation analysis among various indicators of CM66.

	T	SS	SME	A	B_1_	B_2_	B_3_	AM	$\overset{-}{R_{h}}$
A	−0.852 *	−0.509	−0.598						
B1	−0.803	−0.523	−0.602	0.984 **					
B2	−0.785	−0.464	−0.538	0.952 **	0.990 **				
B3	−0.758	0.069	−0.001	0.661	0.696	0.749			
AM	0.844 *	0.422	0.505	−0.972 **	−0.990 **	−0.991 **	−0.789		
$\overset{-}{R_{h}}$	−0.143	−0.901 *	−0.905 *	0.607	0.664	0.645	0.066	−0.573	
Hardness	0.268	−0.776	−0.755	0.121	0.077	−0.022	−0.626	0.060	0.637
Crispness	0.668	−0.509	−0.444	−0.350	−0.374	−0.446	−0.892 *	0.498	0.333
WSI	−0.425	0.842 *	0.790	−0.066	−0.082	−0.030	0.291	−0.004	−0.582
WAI	0.034	−0.933 **	−0.927 **	0.430	0.408	0.328	−0.059	−0.325	0.694
*L**	0.692	−0.687	−0.612	−0.253	−0.235	−0.286	−0.701	0.350	0.491
*a**	−0.465	0.797	0.750	0.019	0.035	0.117	0.644	−0.164	−0.600
*b**	−0.169	0.830*	0.820 *	−0.239	−0.191	−0.087	0.517	0.059	−0.662
Δ*E*	−0.600	0.746	0.682	0.150	0.144	0.208	0.680	−0.267	−0.551
SEI	−0.703	−0.705	−0.776	0.961 **	0.945 **	0.898 *	0.528	−0.909 *	0.751
AD	0.770	0.278	0.347	−0.850 *	−0.909 *	−0.950 **	−0.777	0.930 **	−0.563

* *p* < 0.05, ** *p* < 0.01. T: temperature; SS: screw speed; SME: specific mechanical energy; A: 6 < *X* ≤ 12; B_1_: 12 < *X* ≤ 24; B_2_: 24 < *X* ≤ 36; B_3_: 36 < *X* ≤ 100; AM: *X* > 100; $\overset{-}{R_{h}}$: average hydrodynamic radius; WSI: water solubility index; WAI: water absorption index; *L**: lightness; *a**: red/green chromaticity; *b**: yellow/blue chromaticity; Δ*E*: total color difference; SEI: sectional expansion index; AD: apparent density.

**Table 5 foods-14-01829-t005:** Color, solubility, and water absorption of extruded wheat products with varied wheat variety, temperature, and screw speed.

Sample	WAI	WSI	*L**	*a**	*b**	Δ*E*
ECM66-170-100	1.66 ± 0.07 ^a^	81.30 ± 0.85 ^de^	77.66 ± 0.33 ^b^	2.46 ± 0.06 ^g^	18.08 ± 0.17 ^e^	25.44 ± 0.31 ^g^
ECM66-170-130	1.39 ± 0.08 ^b^	83.60 ± 1.13 ^cd^	75.98 ± 0.49 ^c^	2.98 ± 0.18 ^f^	18.96 ± 0.52 ^d^	27.33 ± 0.71 ^ef^
ECM66-170-160	1.41 ± 0.09 ^b^	83.33 ± 1.01 ^cd^	72.25 ± 0.44 ^e^	4.04 ± 0.09 ^c^	21.14 ± 0.32 ^b^	31.71 ± 0.36 ^c^
ECM66-170-190	0.91 ± 0.07 ^c^	89.23 ± 0.08 ^a^	69.41 ± 0.60 ^f^	4.77 ± 0.17 ^a^	22.36 ± 0.37 ^a^	34.77 ± 0.69 ^b^
ECM66-150-190	1.06 ± 0.13 ^c^	90.40 ± 1.54 ^a^	66.96 ± 0.73 ^g^	4.60 ± 0.17 ^b^	20.87 ± 0.36 ^b^	35.92 ± 0.81 ^a^
ECM66-190-190	1.09 ± 0.14 ^c^	85.60 ± 1.98 ^bc^	76.33 ± 0.23 ^c^	3.23 ± 0.09 ^e^	20.04 ± 0.17 ^c^	27.78 ± 0.25 ^e^
ECM55-170-190	1.02 ± 0.11 ^c^	86.45 ± 1.26 ^b^	74.10 ± 0.36 ^d^	3.71 ± 0.08 ^d^	20.88 ± 0.16 ^b^	30.07 ± 0.35 ^d^
ECM104-170-190	1.65 ± 0.08 ^a^	80.93 ± 0.84 ^e^	76.03 ± 0.14 ^c^	2.97 ± 0.05 ^f^	18.60 ± 0.15 ^d^	27.07 ± 0.09 ^f^
EWW-170-190	0.99 ± 0.03 ^c^	84.72 ± 1.77 ^bc^	82.68 ± 0.26 ^a^	0.80 ± 0.05 ^h^	16.16 ± 0.19 ^f^	20.33 ± 0.32 ^h^

Different letters in a column indicate significant differences (*p* < 0.05). WAI: water absorption index; WSI: water solubility index; *L**: lightness; *a**: red/green chromaticity; *b**: yellow/blue chromaticity; Δ*E*: total color difference.

**Table 6 foods-14-01829-t006:** Correlation analysis among various indicators of different wheat varieties.

	SME	A	B1	B2	B3	AM	$\overset{-}{R_{h}}$	Hardness	Crispness	WSI	WAI
A	−0.840										
B1	−0.870	0.996 **									
B2	−0.899	0.986 *	0.997 **								
B3	−0.928	0.957 *	0.979 *	0.992 **							
AM	0.884	−0.992 **	−0.999 **	−0.999 **	−0.986 *						
$\overset{-}{R_{h}}$	−0.967 *	0.948	0.960 *	0.969 *	0.969 *	−0.964 *					
Hardness	−0.887	0.898	0.886	0.878	0.850	−0.881	0.948				
Crispness	−0.886	0.940	0.929	0.919	0.889	−0.923	0.963 *	0.994 **			
WSI	−0.098	−0.276	−0.189	−0.114	0.008	0.151	−0.112	−0.372	−0.369		
WAI	0.505	−0.055	−0.143	−0.222	−0.342	0.182	−0.288	−0.060	−0.046	−0.895	
*L**	−0.759	33	0.776	0.754	0.705	−0.764	0.843	0.970 *	0.951 *	−0.559	0.135
*a**	0.794	−0.861	−0.834	−0.813	−0.766	0.823	−0.884	−0.985 *	−0.976 *	0.524	−0.104
*b**	0.681	−0.844	−0.803	−0.769	−0.701	0.785	−0.814	−0.941	−0.940	0.662	−0.281
Δ*E*	0.736	−0.819	−0.784	−0.758	−0.703	0.770	−0.835	−0.965 *	−0.951 *	0.595	−0.182
SEI	−0.122	−0.434	−0.370	−0.301	−0.194	0.338	−0.135	−0.204	−0.276	0.731	−0.780
AD	−0.636	0.475	0.451	0.444	0.419	−0.444	0.630	0.813	0.746	−0.435	0.013

* *p* < 0.05, ** *p* < 0.01. SME: specific mechanical energy; A: 6 < *X* ≤ 12; B1: 12 < *X* ≤ 24; B2: 24 < *X* ≤ 36; B3: 36 < *X* ≤ 100; AM: *X* > 100; $\overset{-}{R_{h}}$: average hydrodynamic radius; WSI: water solubility index; WAI: water absorption index; *L**: lightness; *a**: red/green chromaticity; *b**: yellow/blue chromaticity; Δ*E*: total color difference; SEI: sectional expansion index; AD: apparent density.

## Data Availability

The original contributions presented in this study are included in the article/Appendix A. Further inquiries can be directed to the corresponding authors.

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
