# Peer review of "The Importance of Molecular Structure for Textural and Physicochemical Properties of Extruded Wheat Flour"

_foods, 2025, doi:10.3390/foods14101829_

Round 1
Reviewer 1 Report
Comments and Suggestions for Authors
The manuscript addresses an interesting and relevant topic: the relationship between processing conditions, molecular structure, and product quality in cereal-based systems. However, while the application of extrusion processing to cereals (including wheat) and the investigation of starch structure and product features has been widely studied, the novelty of the present work is not sufficiently highlighted. The authors are encouraged to clearly state the innovative aspects of their study in the Introduction section, preferably by including a dedicated paragraph comparing their approach with previous literature.
As regards the experimental plan, the rationale behind the selection of the specific cereal varieties is not clearly explained. Additionally, the chemical composition of the raw materials (including starch, protein, lipid, and ash contents) should be reported. Other important parameters, such as amylose content, damaged starch, and functional properties (e.g., water absorption capacity and water solubility index), should also be included to support the interpretation of the results.
Moreover, it is not clearly stated whether starch was isolated specifically for SEC (Size Exclusion Chromatography) molecular weight distribution analysis—please clarify. The starch isolation yield should be reported. The choice of the isolation method should be justified. In particular, why was protease hydrolysis not used, as commonly done in similar studies?
The rationale behind applying different processing conditions only to CM66 is unclear, please justify this choice. The feasibility of the overnight conditioning at 4 °C and the freeze-drying step from an industrial perspective should be discussed. It is not clear whether the extrusion trials were replicated—please specify.
Table 1 reports standard deviations for torque but not for SME. Please explain this choice and clarify how the replicates were handled in the torque measurements.
Were WAI, WSI, and Color parameters measured on the extrudates in their original form, or were they ground to powder before analysis? Please clarify. If they were measured on the fine powder, this should be taken into account while discussing results and correlation.
The manuscript should include more detail on how hardness and crispness were calculated.
The Results section would benefit from a more in-depth discussion, especially in comparison with previous studies. The Authors are encouraged to contextualize their findings not only within the framework of wheat-based systems, but also by referring to analogous studies on rice, corn, or other cereal grains. When discrepancies with the literature are observed, these should be explicitly acknowledged and discussed, highlighting potential reasons for the differences.
The Conclusion section currently reads more like a summary of the findings. Instead, it should clearly convey the main “take-home message” of the study. Moreover, the authors are encouraged to reflect on both the strengths and the limitations of their work, and to provide insights into possible future developments or applications stemming from their research.
Specific comments:
Line 135: remove the number of replicates because they are reported in section 2.5
Abbreviations used in Tables should be explained.
Line 189: it’s Table 2 not 3
Line 221: avoid expressions as “of interest is that…”
Line 246: why was 170/190 used as “an exemplar”?
Line 253 and Lines 369-371: please, include references
Lines 331-332: What is the explanation of the correlation between color change and hardness and crispness? What is the rationale behind it?
I do not like to divide Tables in a and b.
Reviewer 2 Report
Comments and Suggestions for Authors
The manuscript deals with the extrusion of wheat flours and used readings of size exclusion chromatography and physical analyses to compare the effect of two independent variables, temperature and screw speed compared to isolated wheat starches.
Introduction:
Line 42. The affirmative sentence that was extracted from a reference is not entirely true. Although the sectional expansion was higher in corn expanded extrudates than wheat at certain screw speed, at higher screw speed corn extrudates presented lower sectional expansion than wheat. This shows that findings are relative to the condition. Also, the authors are encouraged to check the following manuscript: CARVALHO, C.W.P.; MITCHELL, J.R. Effect of sugar on the extrusion of maize grits and wheat flour. International Journal of Food Science and Technology, v. 35, n. 6, p. 569-576, 2000, where puffed extrudates of wheat and corn processed at same extrusion conditions , they found that wheat were more susceptible to shear and heat than corn due to mainly in terms of starch structure among other properties.
Line 49. The following sentence should be complemented and a given suggestion is: “…thus low moisture extrusion is favored”, when increase in expansion and crispness is desired.
Line 53. It was appreciated the choice of the reference [10] that is one of the best representation of the effect of thermoplastic extrusion on starch granules (molten starch). In this article the authors did not mention the term starch gelatinization, instead they use molten starch to define the effect of the extrusion of starch at restricted (low) water content. Starch gelatinization requires the presence of enough water combined with heat and some stirring (shear) to provoke gradual loss of the crystal region of amylopectin (birefringence) and then swelling. In such harsh extrusion condition, starch is sheared and broken down as water restriction leads to molten starch. Kindly check the use of the term starch gelatinization.
Line 81. Chemical composition of wheat flour is required as it plays an important role on extrusion cooking (not only amylose and amylopectin content).
Line 85 and 93. Please provide mesh in µm or mm.
Line 94. The extrusion process should follow an experimental design with repetition to combine the effect of screw speed and temperature of the last heating zone on different wheat flours.
Line 92. After the starch extraction, the chemical composition should be carried out in order to check their purity, since were produced various starch types.
Line 98. How many zones have this extruder? Please mention the temperature profile from feed to the die zone. Solid feed rate is missed and is needed to be properly controlled. It is not clear if wheat flour and/or starches were processed in the extruder. Please make it clear. Which criteria were used to choose those screw speeds? They are quite low in order to produce expanded extrudates. Photos and microstructure of the extrudates would be required.
Line 101. What is the source of this equation?
Line 102. What type of mill and setup were used. Sample preparation plays a key role in following the extrusion effect, since it can determine extra starch conversion.
Line 127. Why only SEI was measured? Please check the article [10] that was referenced. Apparent density should be also being measured.
Line 144. Angle Hue is missed.
Line 152. What was the selected load cell? How the extrudates were prepared? Dried? Pre-conditioned in humidity chamber till equilibrium? What was the size? Which methodology the authors followed? It was advisable t use a reference that measures crispness as this attribute was considered in the Introduction section.
Line 164. Torque measurement is not necessary to be considered/discussed since it is used to calculate SME.
Line 166. The difference in SME is too large between the samples. It is not clear that this difference can be only attributed to amylose content (which was not measured in this work). Also, this may not possible to conclude, since it would require an experimental design with repetitions (two independent variables).
Line 183. Table 1. Hardness should be expressed in N. How crispness was calculated (not described in Material and Methods).
Lines 175-177. The authors should try to explain the increase of SME in CM66.
Figure 1. Please also show the effect of screw speed on SEC weight distributions.
Line 230. It is not possible to see the effect of shear as it was not showed in Figure 1. What about low screw speeds of 100 and 150 rpm? As mentioned earlier those screw speed are low and the range is short. Differences would be interesting to discuss if range would cover from 100 to the maximum screw speed reachable by this extruder.
Line 284. The word substance is not appropriate, please consider macromolecules. The values of WAI for the extrudates are lower than usual and compatible to non-extruded (raw) flours. Please recheck the results and compare them with the literature.
Reviewer 3 Report
Comments and Suggestions for Authors
- Could the authors clarify the rationale behind selecting the three wheat varieties used in this study?
- Could the authors justify the choice of 190°C as the extrusion temperature, considering that this temperature may cause flour denaturation and, as noted later in the manuscript, significantly impacts all flour properties?
- The authors are encouraged to include the formulas for amylose and amylopectin calculations (referenced on lines 123 and 124) to enhance clarity and facilitate better understanding.
- Section 3.1 would benefit from improved wording to eliminate repetitions of values and terms, ensuring a more concise and polished presentation.
In Table 1,
- could the authors explain the SME value of 660.24 for sample CM104-170-190, given that its processing temperature was only 170°C, yet it exceeds the SME value for CM66-190-190, which was processed at 190°C?
- I would appreciate the authors’ insights into why the crispness values for formulations CM66-170-100 and CM66-190-190 are nearly identical, despite their processing parameters being significantly different.
- Could the authors explain why the hardness/g value for the CM66-170-100 sample is exceptionally high (7077.85 g)?
- Units of measurement for crispness (Newtons) are missing and should be included for consistency and clarity.
- It is recommended that the authors specify the ideal hardness and crispness ratios for extruded products, emphasizing low hardness and high crispness. The CM104-170-190 variety appears to align most closely with these criteria.
- In Figure 1, the letters "N" and "E" are used to denote non-extruded and extruded products, respectively. To avoid confusion, the authors should either homogenize the terminology throughout the document or remove the letter "E" from the graphs.
- In Table 3, the ΔE value for formulation CM66-170-130 is reported as 27.33. Could the authors explain why this value is notably lower compared to other samples?
- The authors are encouraged to include graphs or diagrams illustrating the relationships between crispness, hardness, and other key parameters. This addition would be highly valuable, as emphasized in section 3.4.
Round 2
Reviewer 2 Report
Comments and Suggestions for Authors
Dear authors, substantial improvements were noticed.
However there are still few concerns.
Line 42. It seems that occured a misunderstanding that must be corrected. The suggested manuscript, in which corn grits and wheat flour were processed in a twin screw extruder, it was found that wheat was prone to greater susceptility (conversion) to heat and shear than corn. Please rephrase the sentence that is linked to the reference [4]. The authors explained, based on DSC data, that wheat (plain flour) melts at lower temperature than corn (corn grits). Please correct this sentence.
In line 43 of the revised version, the reference [5] dis not processed corn to produce ingredients for making tortillas, but instead they used wheat (germinated and germinated plus extrusion). Therefore, whey the authors mentioned "corn-based products can exhibit..." if the work dealt with wheat tortillas? Please careful review the sentence and the refered manuscript.
Lines 179-181. Please change the unit of hardness, crispness texture measurement from "g" to "N", in order to follow the International Standard of Units. Correct the Table 2. As advice, please check the classical manuscript: BOUVIER, J.M.; BONNEVILLE, R.; GOULLIEUX, A. Instrumental methods for the measurement of extrudate crispness. Agro Food Industry Hi-tech, v. 8, n. p. 16-19, 1997.
